# Disorder in M$_{n+1}$AX$_n$ phases at the atomic scale

Chenxu Wang[1,2], Tengfei Yang[2], Cameron L. Tracy [1], Chenyang Lu[3], Hui Zhang[4], Yong-Jie Hu[5], Lumin Wang[3], Liang Qi[5], Lin Gu [6], Qing Huang [7], Jie Zhang[8], Jingyang Wang[8], Jianming Xue[2], Rodney C. Ewing[1] & Yugang Wang[2]

Atomic disordering in materials alters their physical and chemical properties and can subsequently affect their performance. In complex ceramic materials, it is a challenge to understand the nature of structural disordering, due to the difficulty of direct, atomic-scale experimental observations. Here we report the direct imaging of ion irradiation-induced antisite defects in M$_{n+1}$AX$_n$ phases using double C$_S$-corrected scanning transmission electron microscopy and provide compelling evidence of order-to-disorder phase transformations, overturning the conventional view that irradiation causes phase decomposition to binary fcc-structured M$_{n+1}$X$_n$. With the formation of uniformly distributed cation antisite defects and the rearrangement of X anions, disordered solid solution γ-(M$_{n+1}$A)X$_n$ phases are formed at low ion fluences, followed by gradual transitions to solid solution fcc-structured (M$_{n+1}$A)X$_n$ phases. This study provides a comprehensive understanding of the order-to-disorder transformations in M$_{n+1}$AX$_n$ phases and proposes a method for the synthesis of new solid solution (M$_{n+1}$A)X$_n$ phases by tailoring the disorder.

[1] Department of Geological Sciences, Stanford University, Stanford, CA 94305, USA. [2] State Key Laboratory of Nuclear Physics and Technology, Center for Applied Physics and Technology, Peking University, 100871 Beijing, China. [3] Department of Nuclear Engineering and Radiological Sciences, University of Michigan, Ann Arbor, MI 48109, USA. [4] Department of Materials Science and Engineering, Monash University, Clayton, VIC 3800, Australia. [5] Department of Materials Science and Engineering, University of Michigan, Ann Arbor, MI 48109, USA. [6] Institute of Physics, Beijing National Laboratory for Condensed Matter Physics, Chinese Academy of Sciences, Beijing 100190, China. [7] Ningbo Institute of Material Technology and Engineering, Chinese Academy of Sciences, 315201 Ningbo, China. [8] Shenyang National Laboratory for Materials Science, Institute of Metal Research, Chinese Academy of Sciences, 110016 Shenyang, China. Correspondence and requests for materials should be addressed to R.C.E. (email: rewing1@stanford.edu) or to Y.W. (email: ygwang@pku.edu.cn)

Structural disorder in materials can give rise to desirable physical and chemical properties, such as charge transport in high-mobility conjugated polymers[1], thermoelectric effects in graphene and nanostructured carbon materials[2], and magnetoresistance behavior in double perovskites[3]. Ion irradiation is one means of producing such disorder via the production of large numbers of defects[4,5]. Therefore, ion beam irradiation is commonly used to simulate radiation effects in materials used in advanced nuclear energy systems. Ion beam irradiation can also be used to synthesize materials with unique properties[6]. Highly ordered, compositionally complex ceramics are especially prone to disordering under irradiation[7,8]. Understanding the mechanism of the order-to-disorder transformation in such materials is critical to their technological applications.

Ternary $M_{n+1}AX_n$ phases, where M represents an early transition metal, A represents an A-group element, X represents carbon or nitrogen, and $n = 1$, 2, or 3, exhibit highly ordered hexagonal (hex) nano-layered structures ($P6_3/mmc$) consisting of $n$ layers of edge-sharing $M_6X$ octahedra interleaved by close-packed A layers. Owing to this complex characteristic structure, this class of compounds exhibits unique combinations of properties typical of both metals and ceramics, such as easy machinability, high temperature strength, high electrical and thermal conductivities, and excellent oxidation and corrosion resistance[9–11]. Owing to these remarkable properties, $M_{n+1}AX_n$ phases are promising candidate materials for applications involving extremely harsh environments, wherein materials are subjected to high temperatures, chemically reactive surroundings, and intense radiation fields. For example, they have been proposed as coatings on zirconium alloy cladding in advanced nuclear systems to improve accident tolerance[12,13]. Previous studies have investigated the radiation effects of $M_{n+1}AX_n$ phases, including mechanical properties[14], behavior of helium bubbles[15], and swelling[16]. However, the nature and mechanisms of atomic-scale damage resulting from exposure of these materials to radiation have been debated for many years. To date, there are three different explanations for the irradiation-induced transformation to the face-centered cubic (fcc) structures: decomposition into the corresponding binary fcc-structured carbides or nitrides with out-diffusion of the A elements[17]; formation of a new fcc $M_{n+1}(A, X_n)$ composition with A atoms randomly redistributed on the X sublattice[18]; hex-to-fcc transformation due to mixing of both M and A cations onto the sublattice of the other[19]. In the absence of direct observation of local atomic rearrangements accompanying this order–disorder transformation, the precise mechanism cannot be accurately determined, giving rise to these conflicting explanations.

Recently, an advanced imaging technique, aberration-corrected scanning transmission electron microscopy (STEM), has enabled direct atom-by-atom imaging and chemical identification in complex materials[20–22]. The intensity of an atomic column in high-angle annular dark-field (HAADF) STEM images is proportional to $\sim Z^2$ ($Z$ is the atomic number), such that HAADF signals from light atoms are much weaker than those from heavy atoms, allowing for the accurate differentiation of relatively heavy elements[21]. In contrast, the intensity of atoms in annular bright-field (ABF) STEM images exhibits a $Z^{1/3}$ dependency, making ABF highly sensitive to relatively light elements, such as Li and O in the $\beta$-$Li_xIrO_3$ phase[23]. In combination, STEM HAADF and ABF techniques are ideal, complementary methods for characterizing the atomic-scale structural evolution of disordered $M_{n+1}AX_n$ phases, providing detailed information about the variation of cation (M and A) and anion (X) arrangements, respectively.

Here we report the direct observation of irradiation-induced antisite defects in $M_{n+1}AX_n$ phases and chemical disordering using high-resolution (HR) aberration-corrected STEM HAADF

and ABF imaging. An order-to-disorder, hex-to-γ-to-fcc phase transformation leads to the formation of metastable solid solution phases, wherein the M and A atoms occupy a single cation site with the ratio of $(n + 1){:}1$ and the X atoms are located at the anion sites with the occupancy of $n/(n + 2)$. Subsequent characterization by atom probe tomography (APT) shows that the A atoms were randomly distributed in the structure of the solid solution phases. Grazing incidence X-ray diffraction (GIXRD) and first-principle calculations elucidate the precise structural parameters of these disordered phases and further suggest that this unique disordering process yields desirable changes to the materials properties. These results conclusively elucidate the controversial atomic-scale mechanism of the order–to-disorder transformation in $M_{n+1}AX_n$ phases and shows that the introduction of tailored disorder can give rise to superior performance of these materials in advanced nuclear energy systems, thus providing a new means of creating superior $(M_{n+1}A)X_n$ solid solution materials by carefully controlled irradiation conditions.

## Results

**Direct observation of antisite defects**. To investigate irradiation-induced structural modification in $M_{n+1}AX_n$ phases, double $C_s$-corrected STEM was employed to study $Ti_3AlC_2$. This technique allows for direct observation of the structure and provides element-specific information at the atomic scale. As a typical C-based $M_{n+1}AX_n$ phase, the atomic structure of $Ti_3AlC_2$ consists of three layers of edge-sharing $Ti_6C$ octahedra interleaved by close-packed Al layers, with Ti atoms in 2a and 4f, Al atoms in 2b, and C atoms in 4f Wyckoff positions. The inner Ti layer is denoted as Ti(I), while Ti layers adjacent to Al layers are denoted as Ti(II). The stacking sequence of all atoms along [0001] is βC<u>A</u>CβAγ-B<u>A</u>BγA, where the underlined letters refer to Al atoms, the Greek letters to C atoms, and the remainder are Ti atoms.

Figure 1g, p show HR STEM HAADF and ABF images, respectively, of $Ti_3AlC_2$ along [11$\bar{2}$0]. In accordance with the relationship between intensity and $Z$, the Ti layers exhibit brighter contrast as compared with the Al layers in the HAADF image. The intensity profile along the purple line illustrates the ordered Ti-Al arrangement (Fig. 1h). Meanwhile, in the ABF image, C atoms are directly observed at the octahedral interstitial sites between the Ti atoms, which is confirmed by the green line profile (Fig. 1q). HAADF and ABF images along [1$\bar{1}$00] corroborate this ordered distribution of the Ti/Al atoms and the arrangement of the C atoms (Supplementary Fig. 1).

Irradiation of $Ti_3AlC_2$ with 1 MeV Au ions induces slight structural modification at an ion fluence of $3 \times 10^{13}$ cm$^{-2}$, despite retention of the hexagonal structure. The peak damage level induced by ion irradiation is ~0.23 dpa (displacement per atom). Compared with the initial structure (Fig. 1g), the intensity of some Al atomic columns increases, as indicated by the white arrows in Fig. 2a, while that of some Ti atomic columns is attenuated. This indicates the formation of $Ti_{Al}$-$Al_{Ti}$ antisite defects as some Ti atoms are displaced to sites initially occupied by Al atoms, and vice versa. The intensity profile along line 1 in Fig. 2a, which shows the intensity along Al-Ti(II)-Ti(I)-Ti(II)-Al layers, also demonstrates altered intensities, compared with that along the purple line in the pristine sample (Fig. 1g, h). In Fig. 2b, assuming that there exists no Al atoms in column 3 (Ti(I)), the relative Ti proportions in column 1 (Al), 2 (Ti(II)), 4 (Ti(II)), and 5 (Al) are 19.7%, 81.4%, 59.5%, and 55.7%, respectively. This disordering process demonstrates that the Al atoms are more easily replaced by Ti atoms in the Ti(II) layers than in the Ti(I) layers. This experimental finding is consistent with simulation results[24] showing that the $Al_{Ti(II)}$ antisite defect in $Ti_3AlC_2$ exhibits the lowest formation energy (0.74 eV) among all defect

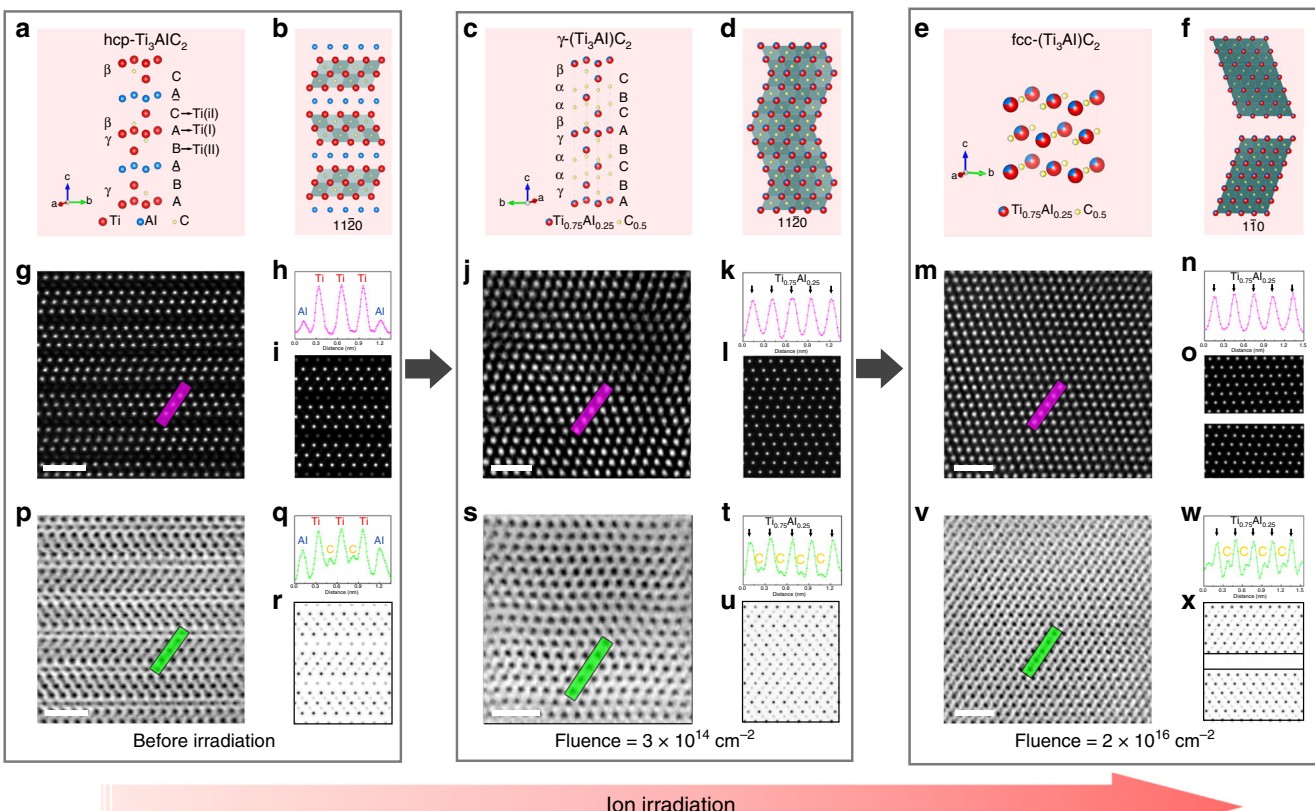

**Fig. 1** Structural models and scanning transmission electron microscopic (STEM) results for Ti$_3$AlC$_2$ before and after irradiation along $[11\bar{2}0]$. Schematic and the corresponding atomic arrangements along $[11\bar{2}0]$ of pristine hex-Ti$_3$AlC$_2$ (**a**, **b**), as well as γ-(Ti$_3$Al)C$_2$ (**c**, **d**) and fcc-(Ti$_3$Al)C$_2$ (**e**, **f**) induced by ion irradiation at the fluence of $3 \times 10^{14}$ and $2 \times 10^{16}$ cm$^{-2}$, respectively. The intermediated Ti layer in **a** is denoted as Ti(I) layer, while Ti layers adjacent to Al layers are denoted as Ti(II) layers. The capital letters and the Greek letters in **a**, **c** represent the stacking sequences of the cations and anions, respectively. The octahedra in **b**, **d**, **f** indicated that the anions are located at the octahedral interstitial sites of the cations. The crystallographic relationship between the hex-(Ti$_3$Al)C$_2$, γ-(Ti$_3$Al)C$_2$, and fcc-(Ti$_3$Al)C$_2$ is $[11\bar{2}0]hex//[1\bar{1}0]fcc$. **g**, **j**, **m** STEM high-angle annular dark-field (HAADF) images of hex-Ti$_3$AlC$_2$, γ-(Ti$_3$Al)C$_2$, and fcc-(Ti$_3$Al)C$_2$. The contrast profiles along the purple lines are shown in **h**, **k**, **n**, which indicates the solid solution process of the Ti/Al atoms at the cation sites. **i**, **l**, **o** Simulated STEM HAADF images of hex-Ti$_3$AlC$_2$, γ-(Ti$_3$Al)C$_2$, and fcc-(Ti$_3$Al)C$_2$, which agree well with the experimental results in **g**, **j**, **m**, respectively. **p**, **s**, **v** STEM ABF images of hex-Ti$_3$AlC$_2$, γ-(Ti$_3$Al)C$_2$, and fcc-(Ti$_3$Al)C$_2$. The contrast profiles along the green lines are shown in **q**, **t**, **w**, which indicates the atomic rearrangements of C atoms at the anion sites. The contrast is inverted for a convenient visualization. **r**, **u**, **x** Simulated STEM ABF images of hex-Ti$_3$AlC$_2$, γ-(Ti$_3$Al)C$_2$, and fcc-(Ti$_3$Al)C$_2$, which agree well with the experimental results in **p**, **s**, **v**, respectively. The scale bars on the HAADF and ABF images correspond to 1 nm

types (1.65 eV for Al$_{Ti(I)}$). The line profile along line 2 (Fig. 2c), which shows the intensity along the initial Al layer, also demonstrates the partial replacement of Al by Ti atoms.

**Hex-to-γ-to-fcc phase transformation.** Although antisite defects were observed in the early stage of irradiation, the hexagonal structure was retained. As further irradiation generates more antisite defects, more extensive effects of disordering on the structure appear. Figure 1j, s show STEM HAADF and ABF images of Ti$_3$AlC$_2$ along $[11\bar{2}0]$ after irradiation to a fluence of $3 \times 10^{14}$ cm$^{-2}$ (~2.3 dpa). After irradiation, the contrast of each atomic column in the HAADF image becomes identical (confirmed by the intensity profile in Fig. 1k), in contrast to the pristine sample. This is attributed to a uniform arrangement of Ti and Al at the cation sites, due to accumulation of disorder in the form of antisite defects. Furthermore, with rearrangement of the cations, the atoms in the initial Al layers move from 2b to 2d sites. Compared with the ABF image of the pristine sample, in which the C atoms exclusively occupy sites between the Ti layers, the ABF image of the irradiated sample (Fig. 1s) shows C atoms located at the octahedral sites between the rearranged cation sites.

Therefore, the stacking sequence of all atoms changes from βC$\underline{A}$Cβ$\underline{A}$γB$\underline{A}$BγA to βCα$\underline{B}$αCβ$\underline{A}$γBα$\underline{C}$αBγA. Ti$_3$AlC$_2$ transforms to a new solid solution phase (denoted as γ-(Ti$_3$Al)C$_2$), where the Ti and Al cations are uniformly distributed with a Ti/Al ratio of 3:1 (in accordance with the material's stoichiometry) and C anions occupy the anion sites with an occupancy of 0.5 (Supplementary Table 2). Together, the rearrangement of the cation and anion atoms leads to variation of the d-spacing between different layers. For example, in the pristine sample, the d-spacing along [0001] between Ti(I) and Ti(II) layers is 2.38(2) Å, while that between Ti(II) and Al layers is 2.27(1) Å. In contrast, in γ-(Ti$_3$Al)C$_2$, the d-spacing between each two cation layers is identical, at 2.41(2) Å, which is larger than any d-spacing between two layers in the pristine sample. This result indicates the presence of swelling along [0001], which results from the accumulation of defects created by the ion irradiation. This is similar to disorder mechanisms previously reported in intermediate hexagonal (Cr,Al)C$_x$[25] and (Ti,Al)N$_x$[26] solid solution phases, in which cations were randomly located at the cation sites.

As the ion fluence increases to $2 \times 10^{16}$ cm$^{-2}$ (~150 dpa), the stacking sequence is further altered to AγBαCβAγBαCβ (Fig. 1e, f), which indicates the transformation of γ-(Ti$_3$Al)C$_2$ to a

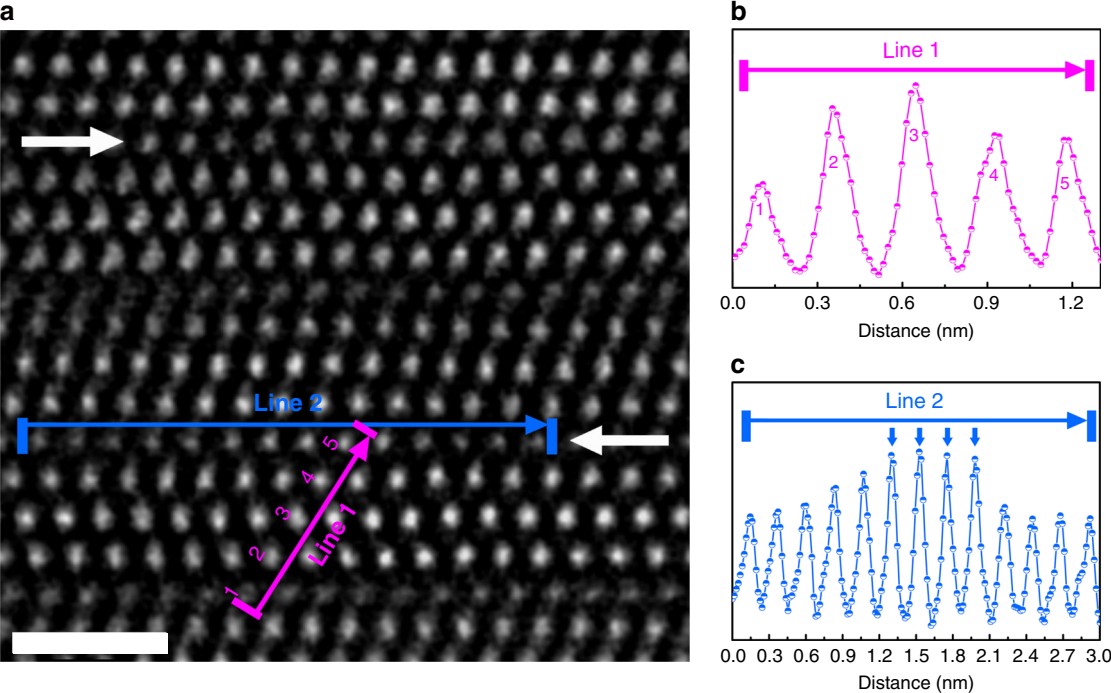

**Fig. 2** Direct observation of cation antisite defect. **a** Scanning transmission electron microscopic high-angle annular dark-field (HAADF) image of $Ti_3AlC_2$ after irradiation at $3 \times 10^{13}$ cm$^{-2}$ along $[11\bar{2}0]$. The white arrows indicate the initial Al layers, whose contrast changed compared to the initial hex-$Ti_3AlC_2$. **b**–**c** Contrast profiles along lines 1 and 2 in **a**, respectively, which directly show the variation of the contrast (indicated by the blue arrows) due to the formation of $Ti_{Al}$-$Al_{Ti}$ antisite defects induced by ion irradiation. The scale bar on the HAADF image correspond to 1 nm

nano-twinned solid solution fcc phase (denoted as fcc-$(Ti_3Al)C_2$, also shown in Supplementary Figs. 1, 2e, j). This phase transformation is triggered by the irradiation-induced formation of stacking faults, which are generated by the dissociation reactions of perfect dislocations in the basal plane[27]. The hex-γ-fcc phase transformation was also observed in the diffraction patterns (Supplementary Fig. 2). Intensity line profiles in HAADF and ABF images demonstrate that the Ti/Al cations and the C anions are uniformly distributed over the cation and anion sites, respectively (Fig. 1m, v). Simulation of the STEM images of all three phases (initial hexagonal $Ti_3AlC_2$ phase, γ-$Ti_3AlC_2$, and fcc-$Ti_3AlC_2$) agree well with the experimental results (Fig. 1i, l, o, r, u, x). Additionally, simulation of their diffraction patterns agree well with experimental results (Supplementary Fig. 3). The $d$-spacing along [111] in the fcc-$Ti_3AlC_2$ increases to 2.45(1) Å due to the continuous accumulation of microstrain.

Once the order–disorder, hex-to-γ-to-fcc phase transformation occurs, the fcc structure persists and the size of the fcc nano-domains continuously increases up to $4 \times 10^{16}$ cm$^{-2}$ (~300 dpa), the highest fluence achieved, suggesting sluggish growth of the fcc phase (Supplementary Fig. 4) and excellent resistance to amorphization despite extensive disordering. This phenomenon is attributed to the presence of high densities of twin boundaries, which strengthen materials[28] and facilitate radiation tolerance via interstitial emission near grain boundaries[29]. STEM images along $[1\bar{1}00]$ (Supplementary Fig. 1) corroborate this structural analysis.

**Formation of solid solutions with chemical disorder**. Irradiation drives order–disorder, hex-to-fcc phase transformations in the representative $M_{n+1}AX_n$ phase $Ti_3AlC_2$. As mentioned above, in both the γ-$Ti_3AlC_2$ phase and the fcc-$Ti_3AlC_2$ phase, the Ti and Al atoms are uniformly distributed over a single cation site with a Ti/Al ratio of 3:1, consistent with their stoichiometry.

Meanwhile, owing to the rearrangement, C atoms occupy the anion sites with an occupancy of 0.5. Therefore, considering the rearrangement of the Ti/Al cations and C anions, both the γ-$Ti_3AlC_2$ phase and the fcc-$Ti_3AlC_2$ phase are considered solid solutions over individual cation and anion sublattices, such that they can be represented as γ-$(Ti_3Al)C_2$ and fcc-$(Ti_3Al)C_2$.

To further confirm the solid solution nature of these phases and to exclude the possibility of irradiation-induced precipitation or decomposition, APT characterization was performed on this material. Figure 3a–c show the distribution of all the elements (i.e., Ti, Al, and C) in the fcc-$(Ti_3Al)C_2$ phase. These individual elements are uniformly distributed on the spatial atom maps, demonstrating that the fcc-$(Ti_3Al)C_2$ phase is a homogeneous solid solution. The Ti/Al ratio remains 3:1 in the fcc-$(Ti_3Al)C_2$ phase, indicating that there is no phase decomposition. The concentration of each element in the fcc-$(Ti_3Al)C_2$ phase as a function of depth exhibits only moderate stochastic fluctuations with no systematic variation, further indicating only random fluctuations in the solid solution of the fcc-$(Ti_3Al)C_2$ phase.

To demonstrate the generality of this order–disorder mechanism to the $M_{n+1}AX_n$ system, seven different $M_{n+1}AX_n$ phases were irradiated and characterized by GIXRD: $Ti_3AlC_2$, $Ti_2AlC$, $Ti_3SiC_2$, $Nb_4AlC_3$, $V_2AlC$, $Ti_4AlN_3$, and $Ti_2AlN$, which belong to the systems Ti-Al-C, Ti-Si-C, V-Al-C, Nb-Al-C, and Ti-Al-N systems. Total electron density of states (DOS) and orbital projected DOS of all $M_{n+1}AX_n$ phases are shown in Supplementary Fig. 5. Irradiation drives similar hex-to-fcc phase transformations in all of these compositions (Fig. 4). Rietveld refinement results of the GIXRD data (Supplementary Fig. 6) are consistent with the (S)TEM results. Based on the GIXRD data, the unit cell parameters, $a$, of the fcc phases were determined (Fig. 5 and Supplementary Table 3). Comparison of these values with those of the associated fcc binary carbides/nitrides further proves that decomposition to these binary phases does not occur. For

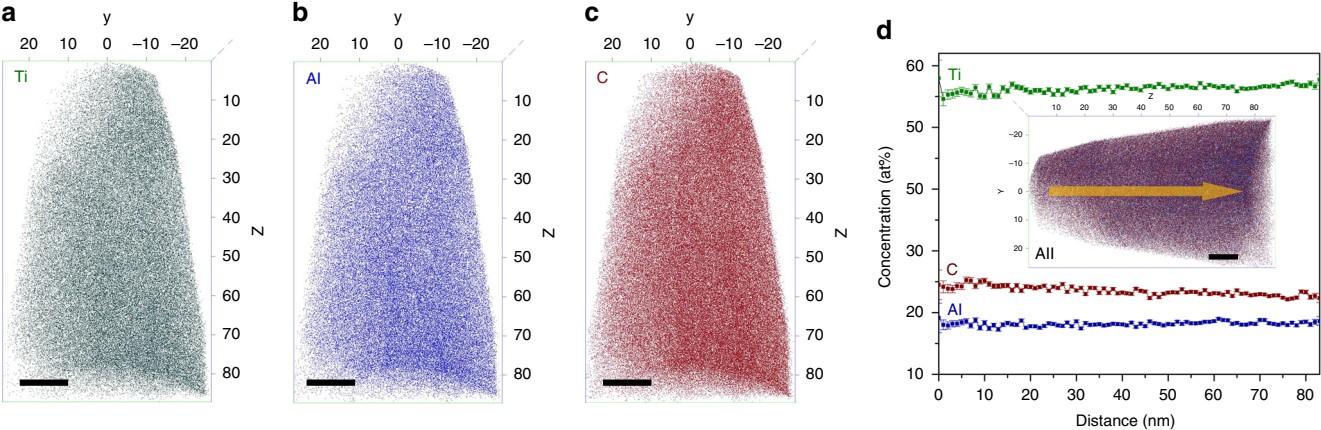

**Fig. 3** Chemical distribution in fcc-$(Ti_3Al)C_2$ solid solution. **a–c** Elemental maps of $Ti_3AlC_2$ after irradiation at $4 \times 10^{16}\,cm^{-2}$ showing homogeneous distribution of Ti, Al, and C elements. **d** Concentrations of these elements as a function of depth, which shows uniformly chemical distribution and proves the existence of Al in the irradiated sample. The scale bars on the APT images correspond to 10 nm

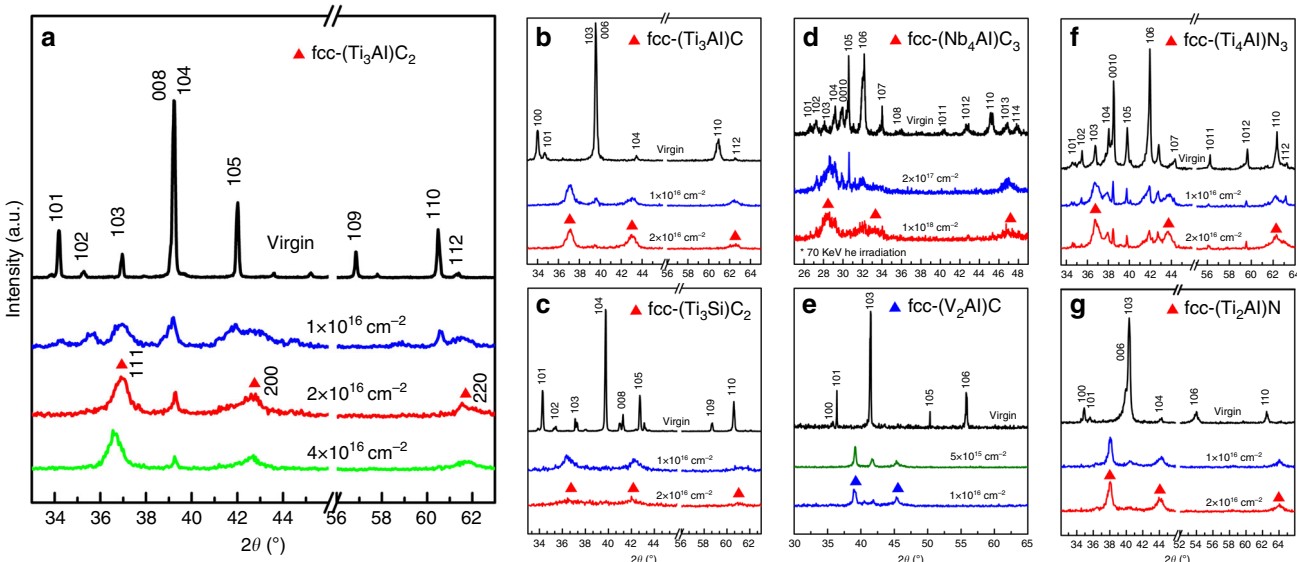

**Fig. 4** Formation of fcc structural solid solutions in seven MAX phases. Grazing incidence X-ray diffraction patterns of $Ti_3AlC_2$ (**a**), $Ti_2AlC$ (**b**), $Ti_3SiC_2$ (**c**), $Nb_4AlC_3$ (**d**), $V_2AlC$ (**e**), $Ti_4AlN_3$ (**f**), and $Ti_2AlN$ (**g**), respectively. $Nb_4AlC_3$ was irradiated with 70 KeV He ions and the rest were irradiated with 1 MeV Au ions. There emerge three new diffraction peaks (111, 200, and 220) characteristic of the fcc phases following irradiation, which are indicated by triangles. This indicates that all of these compounds transform from the initial hexagonal structures to fcc structures. The peak positions of the fcc structures are all slightly different from those of their corresponding binary MX compounds

example, in the Ti-Al-C system, the *a* unit cell parameter of fcc-$(Ti_2Al)C$ and fcc-$(Ti_3Al)C_2$ are both substantially smaller than that of fcc-TiC. The elastic constants and moduli of these fcc-$(M_{n+1}A)X_n$ phases were calculated using first-principles calculations, as shown in Supplementary Table 4. The results indicate that all phases are mechanically stable in the unstressed state in response to perturbation by elastic strains.

There are two main structural differences among these three compounds: the Al occupancy at the cation sites, and the C occupancy at the anion sites. Comparison of the compounds in each system indicate that the unit cell parameter of the fcc-$(M_{n+1}A)X_n$ solid solution decreases with increased A occupancy and X vacancy concentrations. These results agree with those obtained from HRTEM measurements, the SAED patterns, and the ab initio calculations (Fig. 5 and Supplementary Table 3). This compositional trend is attributed to the fact that the Al cation is smaller than Ti, V, and Nb cations in $M_{n+1}AX_n$ phases. Thus

higher Al occupancy results in more severe structural contraction and distortion[30]. Additionally, the lower X occupancy yields higher concentrations of anion vacancies, which further decreases the unit cell parameters.

## Discussion

Previous studies have demonstrated that Al atoms in the hexagonal $Ti_3AlC_2$ phase (A layers in $M_{n+1}AX_n$ phases) are easily displaced from their initial positions along the Al layers[31]. This is because the Ti(II)-Al bond is the weakest among all of the other bonds (such as the strongly covalent Ti-C) in the system, and the Ti(II)-Al-Ti(II) distance is the largest in the $Ti_3AlC_2$ phase. Therefore, it has been found that phase decomposition can occur via Al out-diffusion in $Ti_3AlC_2$ (and some other MAX phases) under some extreme environments, such as high temperatures[11,32], oxygen-enriched environments[33], hydrothermal environments[34], and acidic/alkaline environments[35–37]. $Ti_3AlC_2$ exhibits great thermal

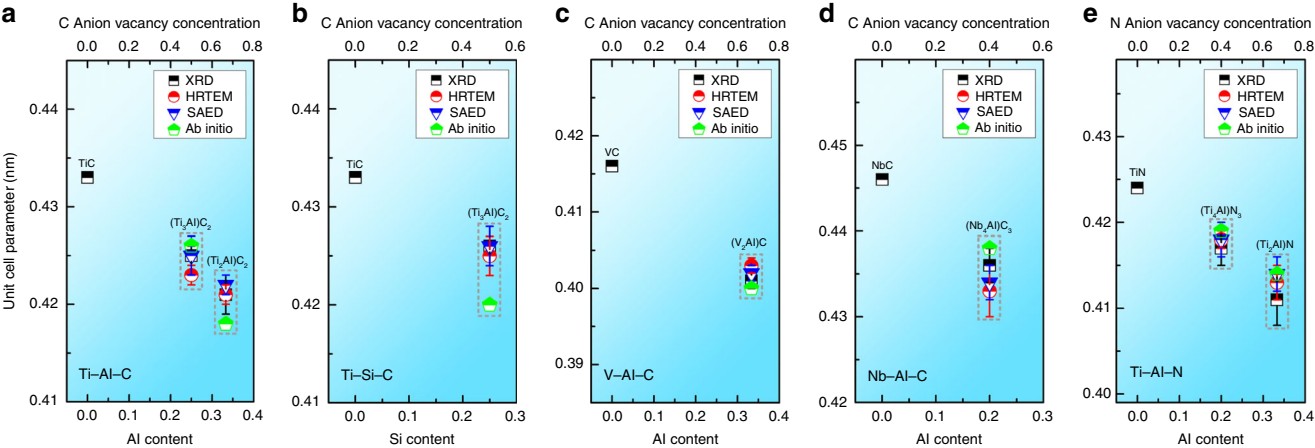

**Fig. 5** Effect of A content and anion vacancy on unit cell parameter of the fcc-$(M_{n+1}A)X_n$ solid solutions. Unit cell parameters of seven different fcc-$(M_{n+1}A)X_n$ solid solutions, i.e., fcc-$(Ti_3Al)C_2$ and fcc-$(Ti_2Al)C$ in Ti-Al-C system (**a**), fcc-$(Ti_3Si)C_2$ in Ti-Si-C system (**b**), fcc-$(V_2Al)C$ in V-Al-C system (**c**), fcc-$(Nb_4Al)C_3$ in Nb-Al-C system (**d**), and fcc-$(Ti_4Al)N_3$ and fcc-$(Ti_2Al)N$ in Ti-Al-N system (**e**), compared to that of their corresponding binary MX compounds, respectively. The error bars represent the standard deviation of unit cell parameter determined from multiple measurements on samples under the same irradiation conditions. The experimental results and the calculation results agree well. The unit cell parameter in each system decreases with both A (Al or Si) content at the cation sites and X (C or N) vacancy concentration. Error bars represent the s.d. of multiple measurements

stability up to around 1300 °C and will not melt congruently at higher temperature but decomposes due to Al out-diffusion instead as shown in the following reaction[11]:

$$Ti_3AlC_2 \rightarrow Ti_3C_2 + Al\,(in\,vacuum,\,T > 1300°C) \quad (1)$$

The remaining twinned $Ti_3C_2$ slabs can be detwinned and recrystallize to (111)-oriented $TiC_{0.67}$ layers under annealing, accompanied by the formation of pores in the material. This phase decomposition process has been utilized to synthesize new materials, such as noble-metal-containing transition-metal carbides/nitrides[38] and two-dimensional MXene nanosheets[35,36].

Based on the STEM, APT, and GIXRD results in this study, $Ti_3AlC_2$ transforms from the initial hexagonal phase to an fcc phase in response to ion irradiation. Some prior studies have claimed that $M_{n+1}AX_n$ phases decompose to binary fcc-structured TiC under irradiation[17]. Yet the results reported here clearly demonstrate that the irradiation-induced hex-to-fcc transformation process is distinct from that which occurs under other extreme environments, as it does not involve decomposition and instead produces a distinct, highly disordered solid solution fcc phase. The previous, although erroneous, attribution of this transformation to a decomposition process is understandable because the disordered fcc-structured phases produced by irradiation are crystallographically similar to TiC, especially when considering the structural distortion induced by irradiation that hinders determination of the phase based on its unit cell parameters. However, in this study, STEM imaging and APT reconstructions clearly show that Ti, Al, and C atoms are uniformly distributed in a solid solution in the irradiation-induced fcc-structured phase, such that the atomic ratio between Ti and Al atoms remains 3:1, as in the initial hexagonal $Ti_3AlC_2$ phase. This indicates that ion irradiation drives an otherwise unachievable phase transformation to an fcc solid solution phase instead of phase decomposition, which is the response of this system to most extreme environments. Improved understanding of the phase modification mechanism demonstrates the manner in which ion irradiation can be used as a processing tool to uniquely tailor the structure of $M_{n+1}AX_n$ phases.

The formation of the irradiation-induced fcc solid solution phases, fcc-$(M_{n+1}A)X_n$, in all five M-A-X systems studied here is

attributed to the production of $M_A$-$A_M$ antisite defects at the early stage of irradiation (low ion fluence). The accumulation of these antisite defects leads to chemical disorder at the cation sites, accompanied by the rearrangement of the X anions according to the following order–disorder defect reactions:

$$M_M + A_A \rightarrow M_A + A_M\,(cation\,antisite\,formation) \quad (2)$$

$$X_X \rightarrow V_X + X_i\,(X\,anion\,Frenkel\,formation) \quad (3)$$

Related chemical disordering processes have been observed in some complex oxides, such as $A_2B_2O_7$ compositions with pyrochlore structure, under ion irradiation[7,8,39]. These compounds transform to chemically disordered, fcc-structured $(A_2B_2)O_7$ solid solutions by the formation of $A_B$ and $B_A$ cation antisite defects and the accompanying rearrangement of anions. This disordering process strongly influences many transport properties (e.g., thermal conductivity), as well as mechanical and magnetic properties[40,41]. Therefore, it is expected that the phase transformation from the initial hexagonal $Ti_3AlC_2$ phase to the metastable $\gamma$-$(Ti_3Al)C_2$ similarly enhances the radiation tolerance of this material, allowing it to incorporate a high concentration of irradiation-induced defects while maintaining crystallinity. Additionally, it might prove useful as a means of tailoring properties of materials in this system.

Consistent with the role of chemical disordering and antisite defect formation, the radiation tolerance of $M_{n+1}AX_n$ materials arises from irradiation-activated stacking fault and twin boundary processes. As the ion fluence increases, large numbers of stacking faults are introduced by the dissociation reactions of the perfect $\frac{1}{3}\langle 11\bar{2}0\rangle(0001)$ dislocations that are the most energetically favorable[42]. This triggers the phase transformation from the $\gamma$-$(Ti_3Al)C_2$ phase to a rocksalt-like fcc-$(Ti_3Al)C_2$ solid solution phase by changing the stacking sequences. The Al atoms in the fcc-$(Ti_3Al)C_2$ phase stabilize the nano-twinned structure[43,44]. This phenomenon suppresses the growth of the fcc-$(Ti_3Al)C_2$ nano grains from $2 \times 10^{16}$ cm$^{-2}$ to $4 \times 10^{16}$ cm$^{-2}$, which results in the excellent amorphization resistance of these materials. The structure of the initial $Ti_3AlC_2$ phase can be described as nano-twinned $TiC_{0.67}$ interleaved by Al layers, where Al layers can be

also considered as twin boundaries in the unit cell. High densities of twin boundaries have been shown to strengthen materials[28,45] and facilitate radiation tolerance[29]. Bai et al.[29] investigated the interactions between irradiation-induced defects and grain boundaries in twinned copper, finding that the grain boundaries could both absorb interstitials formed by atomic displacement and emit them to annihilate vacancies in the surrounding volume, leading to dramatic reduction in the extent of radiation damage[46] and enhancement of the radiation tolerance of the materials. In all of the initial hexagonal phases, the γ-(Ti₃Al)C₂ phase, and the fcc-(Ti₃Al)C₂ phase with nano-twinned structures, the existence of plentiful "twin boundaries" retards the accumulation of the irradiation-induced defects, which mitigates degradation of the physical and mechanical properties of these materials.

Additionally, in conventional $M_{n+1}AX_n$ phases, there is only a single element occupying each of the M, A, and X sites. More recently, several quaternary MAX-derivative phases have been synthesized, including ordered quaternary phases $Cr_2TiAlC_2$[47] and $Mo_2Ti_2AlC_3$[48], as well as random solid solutions $Ti_3(Al_{1-x}Si_x)C_2$[49] and $Ti_2Al(C_xN_{1-x})_y$[50,51]. These phases possess many properties distinct from those of the corresponding ternary compounds (e.g., improved strength and hardness). Similarly, both the irradiation-induced γ-(Ti₃Al)C₂ phase and the fcc-(Ti₃Al)C₂ phase synthesized in this work exhibit $(M_{n+1}A)X_n$ solid solution structures, suggesting that they may possess similarly improved properties without the need for the incorporation of additional elements. To our knowledge, these new solid solution phases have not been previously synthesized by any other means of processing. The existence of these solid solutions tremendously expands the accessible phase space in the $M_{n+1}AX_n$ family of compositions and provides the possibility of "tuning" the properties of these materials by precisely tailoring of the disordered phase fraction of the hex-γ-fcc mixture.

In summary, we have directly observed the formation of cation antisite defects in Ti₃AlC₂ by ion irradiation and investigated the polymorphic hex-to-γ-to-fcc phase transitions. The metastable γ-(Ti₃Al)C₂ and the fcc-(Ti₃Al)C₂ phases are identified as solid solutions in which Ti and Al atoms randomly occupy the cation sites and C atoms are located at the anion sites with the occupancy of 0.5, indicating an ion irradiation-induced order–to-disorder process. These findings are confirmed by the contrast variation in STEM (HAADF and ABF) images and elemental distributions in APT reconstructions, disproving previous reports of phase decomposition to binary TiC as a response of these materials to ion irradiation. Understanding the transition mechanism in Ti₃AlC₂ plays a key role in studying the radiation effects of the class of $M_{n+1}AX_n$ phases and improving their applications under extreme radiation environments. We have also shown, for all the $M_{n+1}AX_n$ phases in this study, that ion irradiation drives phase transformations to otherwise unachievable solid solution phases, thus providing a new strategy for the design of new derivatives of the $M_{n+1}AX_n$ phases with tailored disorder and potentially improved properties within the typically thin ion–solid interaction region.

## Methods

**Material synthesis**. Polycrystalline $M_{n+1}AX_n$ samples used in this study were synthesized by hot isostatic pressing at Ningbo Institute of Materials Technology and Engineering and Shenyang National Laboratory for Materials Science. Elemental powders in stoichiometric proportions were mixed and pressed in a graphite mold and then hot-pressed in a flowing Ar atmosphere. Details of the synthesis process have been published elsewhere[10]. All samples were polished with diamond paste suspensions and washed with acetone prior to irradiation.

**Ion irradiation**. Irradiation of the well-polished $M_{n+1}AX_n$ samples with 1 MeV Au⁺ ions at room temperature was carried out with a $2 \times 1.7$ MV ion accelerator at Peking University. Samples were irradiated to a series of fluences ranging from $1 \times$ $10^{14}$ to $4 \times 10^{16}$ cm⁻², with the beam current held below 1 μA cm⁻² in order to avoid significant bulk heating. Damage profiles and the implanted ion concentration as a function of depth induced by 1 MeV Au⁺ ion irradiation at $1 \times 10^{16}$ cm⁻² in Ti₃AlC₂ were calculated using the SRIM code[52], as shown in Supplementary Fig. 7.

**GIXRD measurement**. Synchrotron GIXRD measurements were performed at beamline 1W1A of the Beijing Synchrotron Radiation Facility, with a wavelength of 0.1547 nm. Diffraction patterns were measured using a NaI scintillation detector with a 2θ interval of 0.05° and an incident angle of 0.5°, such that the scattering depth in the GIXRD investigation was ~200 nm. This is roughly consistent with the ion damage peak, so as to minimize signal from the unirradiated area in the diffraction patterns.

**STEM characterization and simulation**. Cross-sectional samples for STEM observations were mechanically polished, then ion milled for sufficient electron transparency. The atomic structures of the samples before and after irradiation were characterized using a double $C_S$-corrected JEOL JEM-ARM200F S/TEM operated at 200 kV with a STEM-HAADF resolution of 78 pm. STEM-ABF and -HAADF images were obtained at 11–22 and 90–250 mrad, respectively. When performing the STEM experiments, we focused on the peak damage region, such that the region containing the maximum concentration of the deposited Au ions was avoided. The thickness of all samples were determined using the DigitalMicrograph software based on EELS spectrum. STEM images were simulated using the QSTEM software[53], which is based on a multislice algorithm. In the beam direction, the structure was divided into slices with an approximately equal thickness of 1 Å. To account for the thermal diffuse scattering, the frozen phonon method was used and the results were averaged over 30 frozen phonon configurations. Microscope characteristic parameters of JEM-ARM200F were used. The collection angular range of the HAADF and ABF detectors was fixed at the same values as were used in the experimental measurements.

**APT characterization**. Needle-shaped tips were prepared using a focused ion beam system (FEI Nova 200/Zeiss Auriga) to dimensions of ~20 × 20 × 200 nm³ for APT analysis. APT measurements were carried out using a local electrode atom probe (CAMECA LEAP 4000×) in pulse laser mode. A laser pulse of 50 pJ energy and 200 kHz frequency was used, while the specimen temperature was kept at 30 K. The CEMECA IVAS 3.6.12 software package was used for data reconstruction and analysis.

**First-principles calculation**. First-principles calculations based on density functional theory were performed using the Vienna Ab-initio Simulation Package (VASP)[54]. The projector augmented wave method[55] and the exchange-correlation functional depicted by the generalized gradient approximation by Perdew, Burke, and Ernzerhof[56] were employed. The plane wave energy cutoff was set at 500 eV to ensure the accuracy of the calculations. The energy convergence criterion of the electronic self-consistency was set at $10^{-6}$ eV per atom for all calculations. To mimic the disorder-mixing at individual sublattice sites, the simulation supercells were constructed using the alloy theoretic automated toolkit[57] based on the special quasi-random structure method[57,58]. The reciprocal $k$-point-meshes for all the first-principles calculations are generated using the automatic $k$-mesh generation scheme implemented in VASP with a length of 40. The supercell structures were relaxed by implementing the Methfessel–Paxton method[59] to obtain the equilibrium lattice parameters at 0 K.

## Data availability
The data that support the findings of this study are available from the corresponding author on request.

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

## Acknowledgements

This work was financially supported by the National Magnetic Confinement Fusion Energy Research Project of China (2015GB113000), the National Natural Science Foundation of China (11675005). C.L.T. and R.C.E. were supported by the Energy Frontier Research Center "Materials Science of Actinides" funded by the U.S. Department of Energy (DOE), Office of Science, Office of Basic Energy Sciences (Grant No. DE-SC0001089). Y.J.H. and L.Q. acknowledge support by startup funding from the University of Michigan. This research was supported in part through computational resources and services provided by Advanced Research Computing at the University of Michigan, Ann Arbor. This research used resources of the National Energy Research Scientific Computing Center, a DOE Office of Science User Facility supported by the Office of Science of the U.S. Department of Energy under Contract No. DE-AC02-05CH11231. Part of this work was performed at the Stanford Nano Shared Facilities (SNSF), supported by the National Science Foundation under award ECCS-1542152. The authors thank the staff of Beijing Synchrotron Radiation Facility (BSRF) for the GIXRD measurements.

## Author contributions

C.W. and Y.W. conceived of the research strategy and designed the experiments; Q.H., J.Z. and J.W. synthesized the bulk samples; C.W., C.L., L.G. and L.W. carried out the ion irradiation and collected the TEM data; H.Z. performed the STEM simulation; Y.-J.H. and L.Q. performed the ab initio calculations; C.W., T.Y., J.X. and C.L.T. analyzed the data; C.W., C.L.T. and R.C.E. prepared the manuscript. All authors discussed the results and contributed to the manuscript.

## Additional information

**Competing interests:** The authors declare no competing interests.

