## [Peer Review File · Nature Communications]

Reviewers' comments:

Reviewer #1 (Remarks to the Author):

The authors present a comprehensive study of irradiation-induced disorder in a large (and representative) number of $Mn_{n+1}AX_n$ phases. These results are potentially of very high importance to the field and explain many of the microscopic details and phase transformations occurring during irradiation of this class of materials. This is one of the most important possible application areas for MAX phases, with great strides being taken toward using these materials in next-generation reactors. The (S)TEM results, including chemical analysis, are indeed beautiful, and complete. Therefore, the work is both of very high importance to the field, comes a long way in support of its conclusion (though some additional work is required) and potentially constitutes an advance in our understanding of these materials that is likely to influence thinking among researchers in the field. With some additional work, it is in my view likely that this paper will reach the level required for publication in Nature Communications.

Overall comments:

1. As mentioned, the (S)TEM results, including chemical analysis, are beautiful indeed, and complete. Nonetheless, the question arises: how representative are the results macroscopically? The volume probed is extremely limited. Normally, for these type of structure determinations, the structural model proposed from TEM is verified by Rietveld refinements of XRD data (or neutron diffraction), and the Rietveld refinement is presented in complete detail. The synchrotron XRD data appear underused in the paper, and the authors should proceed with additional analysis to confirm the structures from these macroscopic data (or at least show that they are consistent). The authors do list lattice parameters determined from XRD in the Supplementary info, but it is not clear how this was done. Either way, the authors need present a complete case that the determined structures in TEM are macroscopically representative.
2. The synchrotron XRD results should be in the main paper, not the Supplementary information. In contrast, the atom-probe results do not provide too much information, only that the samples are homogeneous on this length scale. This could be just one sentence in the main paper and the APT results moved to the SI.
3. The Supplementary Information needs explanatory text to describe and present all results in full detail, not only captions for figures and tables.
4. The DFT parts of the paper requires additional work and more complete presentation. To start with, all DFT calculations and results should be presented in full. Right now, only lattice parameters of the structures are stated, but it is not fully clear how these were determined. What were the relaxation criteria?

Furthermore, the structures should be characterized more by DFT:

- a) thermodynamic stability relative to each other? (we already know that all these MAX phases are thermodynamically stable relative to any known competing phases, so there is no need to do that calculation – it is sufficient to compare the structures with each other)
- b) Electronic band structures and phonon dispersion should be calculated and discussed in detail.
- c) For the application in mind, it would probably be interesting (and rather straightforward) to calculate also elastic constants.

In summary, additional computational work is required, and the results should be presented in full (mainly in the SI). For an example of the level of presentation needed, a good example is the

Nature Materials paper by Fashandi et al, ref 36 in the present paper.

Questions/clarifications on the structure determinations:

5. The starting structure is not hcp. In the MAX structure, only the MX layers are close-packed. The MAX structure is hexagonal, but not hexagonal close-packed. Please correct everywhere.

6. How is it possible that the structure can remain stable even with 67% C vacancies?

7. For the intermediate step, the "gamma" structure should be more clearly explained in words in the text, and a complete description/structural model of the "gamma" structure should (if possible) be provided in the Supplementary Info. The starting MAX structures (from literature) and the resulting fcc structures are provided in the Supplementary information, but the gamma structure is not.

8. Similar disorder occurs in hexagonal intermediate phases (Cr,Al)_x, (Ti,Al)_{Nx}. These appear similar to the intermediate gamma phase observed here and should be mentioned and discussed: Abdulkadhim, ..., Schneider Surf. Coat. Technol. 206 599 2011 [hexagonal (Cr,Al)₂C_x] Cabioch, et al, Mater. Res. Bull. 80, 58, 2016 [hcp-like (Ti,Al)_N solid solution]

Other (minor) comments:

9. In the discussion on solid solutions on line 339 and onwards, please make the distinction better between ordered quaternary phases and random solid solutions. (Cr₂Ti)AlC₂ is a quaternary phase – or ordered solid solution depending on how you view it - with a nominally fixed stoichiometry [see also B. Anasori et al, Journal of Applied Physics 118:94304-7 (2015) for a more detailed discussion on this phenomenon]. The other example are regular solid solution, which is a very different thing. Also, a detail for the carbonitrides: probably a better (=more original from the same authors) reference than ref 47 is Cabioch et al J. Eur. Ceram. Soc. 32 1803 2012, and the original is M.W. Barsoum, M. Ali, T. El-Raghy Metall Mater Trans A, 31 (2000), pp. 1857-1865

10. There are no CoI or Data Availability Statements. For the latter, it is highly recommended that the authors make all data and files available in a public data repository (eg Zenodo) after acceptance.

11. Abstract: "A grand challenge in complex ceramic materials is to understand the nature of their structural disordering" Please correct to "A challenge...".

A "grand challenge" is an established concept; it is much more than a research problem, it is a major undertaking with global impact, something like for example stopping global warming. (en.wikipedia.org/wiki/Grand_Challenges)

12. Materials and methods: "Details of the synthesis progress have been published elsewhere." Please provide references.

13. The paper is well written but should be carefully proofread for spelling and mixed up words. For example, lines 134-137: "...edge-sharing Ti₆C octahedral (SHOULD BE "OCTAHEDRA", NO L) interleaved by close-packed Al layers, with Ti atoms in 2a and 4f, Al atoms in 2b, and C atoms in 4f Wyckoff positions. The intermediated (SHOULD BE "INNER") Ti layer is donated (SHOULD BE "DENOTED") as Ti(I) layer (REMOVE "LAYER"), while Ti layers..." Etc.

*Line 95: "In the absent direct observation..." should be "In the absence of direct observation..."

*The title should rather be "Disorder..." than "Disordering..."

Reviewer #2 (Remarks to the Author):

Due to their remarkable properties, MAX phases are promising candidate for applications in extremely harsh environments (high temperature, chemically reactive surrounding and intense radiative environments). For instance, they have been proposed as coatings on zirconium alloy cladding in advanced nuclear systems to improve accident tolerance (see for example European project ID 740415 and references 12 and 13 in the manuscript). The nature and mechanisms of atomic scale damage resulting from MAX phase exposure to radiation is still debated. Moreover, understanding the nature of the structural disordering of complex ceramic materials is a challenge.

The authors report the direct observation of irradiation-induced antisite defects in MAX phases and chemical disorder using high resolution aberration-corrected STEM HAADF and ABF imaging. The authors provide compelling evidence of order-to-disorder phase transformations; overturning the conventional view that irradiation causes phase decomposition to binary carbide or nitride. From convincing TEM images and analyses, from Atom Probe Tomography and from Grazing Incident X-ray Diffraction, it is demonstrating that a uniform distribution of cation antisite defects and the rearrangement of X anions is formed. It led to a γ -(Mn+1A)X_n solid solution at low fluence whereas a fcc-structured (Mn+1A)X_n solid solution is formed at higher fluence. The manuscript is well written and references are well chosen. The originality of the approach (the use of HAADF and ABF techniques to image the disorder) is clearly explained and is convincing. To my opinion, the results are of interest for the MAX phase community and also for researchers that are working on order-disorder transition and for nuclear field.

Finally, I think that the paper can be accepted for publication.

I have nevertheless some comments in order to slightly improve the manuscript:

- 1 - page 6 line 116-117: I do not understand the (n+1)/n ratio ; the proportion of M atoms on the cation site should be (n+1)/(n+2). In the manuscript, the anion site occupancy is 50% (I agree with this value) whereas the value given in line 117 is n/(2n+1).
- 2 - page 7 line 136: Change "donated" by denoted.
- 3 - page 8 line 161-164: There is, to my opinion, a mistake. AlTi(II) antisite defects in Ti₃AlC₂ exhibits the lowest formation energy (0.74 eV) among all defect types (1.65 eV for Al Ti(I) which is indeed consistent with your results.
- 4 - It is quite important to have an idea of the sample thickness and to know if:
 - the different samples have about the same thickness
 - the correct thickness has been used in simulated STEM HAADF images.
- 5 - Page 13 line 262-264: The sentence "This is attributed tophase." has to be rewritten.
- 6 - The authors mention, in the conclusion (page 17 lines 364-367), that ion irradiation allows phase transformations to otherwise unachievable solid solution phases. It can lead to new derivatives of the MAX phases with tailored disorder and potentially improved properties. It has to be moderated as the size of the solid solution sample is necessary very small (typically the size of the damage region which should be about 400 nm for irradiation performed with Au⁺ at 1 MeV).
- 7 - The reader (and the referee) would appreciate to know if the quality of the STEM HAADF and ABF images was as good as the presented ones for a lot of sample regions. Indeed, irradiation introduces damages and it is very often quite difficult to get good quality images due to this damages.
- 8 - The damage profile should be given in order to get an idea of the depth of the damage region.
- 9 - It would be of interest to get the displacement per atom (dpa) for the different fluences that have been used and to have an idea of the dpa at the observed region (figure 1g, 1j and 1m).
- 10 - Page 23 line 467: change "donated" by denoted
- 11 - Figure 1j and 1s are of quite poor quality. Is the sample thickness higher than the ones that allows obtaining the other images (1g, 1p and 1m, 1s) ? If it is not the case, is it the result of irradiation?

12 - I am surprised that Figure 2 represents simulated electron diffraction patterns. I would say that it is experimental diffraction patterns.

Reviewer #3 (Remarks to the Author):

In this Manuscript, the authors investigated the phase transforms of $M_{n+1}AX_n$ phases under Au^+ ion irradiation process using CS-corrected scanning transmission electron microscopy. The transformation of $M_{n+1}AX_n$ phase under service conditions is an important subject, which is still controversial at present. It seems that different types of irradiated ions produce different results. Although the research in this paper is limited to the Au^+ ion irradiation, it still provides very important information for understanding the phase transition process of $M_{n+1}AX_n$ phase under other irradiation conditions, such as neutron, He^+ and so on.

1. Have the authors evaluated the distribution of injected Au^+ ions in samples and their role in stabilizing the fcc-structured ($M_{n+1}AX_n$) phases?
2. In electron microscopy, electron beam irradiation usually introduces defects in specimen and may lead to local ordered - disordered transformation. Did the authors evaluate the effects of electron irradiation during STEM measurements?
3. Perfect electron microscopy work, it's impressive.

Anyway, it is a valuable work; I'm looking forward to it being published as soon as possible.

Responses to reviewer:

Thank you very much for the kind and detailed comments that are highly important to our research. Please find the answers to the questions as follows:

Reviewer #1 (Remarks to the Author):

1. Question: *As mentioned, the (S)TEM results, including chemical analysis, are beautiful indeed, and complete. Nonetheless, the question arises: how representative are the results macroscopically? The volume probed is extremely limited. Normally, for these type of structure determinations, the structural model proposed from TEM is verified by Rietveld refinements of XRD data (or neutron diffraction), and the Rietveld refinement is presented in complete detail. The synchrotron XRD data appear underused in the paper, and the authors should proceed with additional analysis to confirm the structures from these macroscopic data (or at least show that they are consistent). The authors do list lattice parameters determined from XRD in the Supplementary info, but it is not clear how this was done. Either way, the authors need to present a complete case that the determined structures in TEM are macroscopically representative.*

Answer: Thank you for the suggestions. The representativeness of our characterizations is indeed a primary concern. This was a major motivation for the use of various characterization methods, including (S)TEM, APT and XRD. While (S)TEM and APT measurements were used to characterize the atomic structure and chemical composition at small scales, GIXRD was employed to characterize the overall structural properties. The structural characteristics of the as-irradiated MAX phases described here were derived from the combination of the three characterization results. For the GIXRD experiments, the unit cell parameters were determined from the index and the position of Bragg peaks using the software Unitcell (Holland *et al.* Mineralogical Magazine 61.1 (1997): 65-77.). This has been better explained in the revised manuscript. We also performed Rietveld refinements of the XRD data, as suggested. The figure below, included in the revised manuscript, shows the refinement of XRD data of Ti_3AlC_2 irradiated to an ion fluence of $2 \times 10^{16} \text{ cm}^{-2}$. The phase fractions of γ phase and fcc phase determined from this refinement are 19.1% and 80.9%, respectively, indicating that the fcc phase is dominant at $2 \times 10^{16} \text{ cm}^{-2}$. Additionally, the a unit cell parameter determined from the refinement is 0.424 nm. These representative results of Rietveld refinements, which are consistent with those obtained from the other compositions studied, are consistent with the TEM and APT results shown in our study.

Rietveld refinement of XRD data of Ti_3AlC_2 irradiated to an ion fluence of $2 \times 10^{16} \text{ cm}^{-2}$. The black line, red line and green lines at the bottom, represent the observed data, calculated result and the difference between the two, respectively.

2. Question: *The synchrotron XRD results should be in the main paper, not the Supplementary information. In contrast, the atom-probe results do not provide too much information, only that the samples are homogeneous on this length scale. This could be just one sentence in the main paper and the APT results moved to the SI.*

Answer: The XRD results have been moved to the main paper. Because the APT results demonstrate conclusively the homogeneity of the solid solutions, we prefer to keep them in the main body of the manuscript, and have edited the figures so as to include both GIXRD and APT results.

3. Question: *The Supplementary Information needs explanatory text to describe and present all results in full detail, not only captions for figures and tables.*

Answer: The text in the revised supplementary information has been substantially expanded to fully describe the results presented and the methods by which these results were obtained.

4. Question: *The DFT parts of the paper requires additional work and more complete presentation. To start with, all DFT calculations and results should be presented in full. Right now, only lattice parameters of the structures are stated, but it is not fully clear how these were determined. What were the relaxation criteria? Furthermore, the structures should be characterized more by DFT:*

a) thermodynamic stability relative to each other? (we already know that all these MAX phases are thermodynamically stable relative to any known competing phases, so there is no need to do that calculation – it is sufficient to compare the structures with each other)

b) Electronic band structures and phonon dispersion should be calculated and discussed in detail.

c) For the application in mind, it would probably be interesting (and rather straightforward) to calculate also elastic constants.

In summary, additional computational work is required, and the results should be presented in full (mainly in the SI). For an example of the level of presentation needed, a good example is the Nature Materials paper by Fashandi et al, ref 36 in the present paper.

Answer: In the present work, the simulation supercell is generated by using the Alloy Theoretic Automated Toolkit (ATAT) based on the special quasirandom structure (SQS) method (Zunger, *et al.*, *Physical Review Letters* **65**, 353 (1990); Van De Walle, *et al. Calphad: Computer Coupling of Phase Diagrams and Thermochemistry* **42**, 13–18 (2013).) to mimic the disorder-mixing between anion and vacancies. The SQS method provides a satisfactory periodic supercell approximation to the true random solution phase under the constraint of a computationally tractable cell size. It should be noted that the generated supercell for SQS usually has a monoclinic or triclinic symmetry (Marker, *et al. Computational Materials Science* **142**, 215–226 (2018); Liu, *et al. PloS one* **10**, e0121386 (2015); Shin, *et al. Physical Review B* **76**, 144204 (2007); Jiang, *et al. Physical Review B* **69**, 214202 (2004).), but the high symmetry (i.e. fcc in the present work) is preserved when all atomic sites for disorder-mixing are occupied a single type of species (i.e. anion/vacancy in the present work).

In the present work, the lattice parameters of the fcc structures of a MAX phase are derived from the average atomic volumes obtained from the DFT-relaxed SQS. The relaxations of the supercells in the present work follow a routine scheme which is commonly used for the relaxation of SQS in first-principles calculations (Marker, *et al. Computational Materials Science* **142**, 215–226 (2018); Liu, *et al. PloS one* **10**, e0121386 (2015); Shin, *et al. Physical Review B* **76**, 144204 (2007); Jiang, *et al. Physical Review B* **69**, 214202 (2004).). Specifically, three different relaxation schemes were tested: the first relaxation scheme allowed a complete relaxation of the structure, while the second and third relaxation schemes constrained the relaxation of the ions and cell shape. The structures with the lowest energy that still retain the fcc symmetry were used for the determination of the lattice parameter.

a) thermodynamic stability relative to each other? (we already know that all these MAX phases are thermodynamically stable relative to any known competing phases, so there is no need to do that calculation – it is sufficient to compare the structures with each other)

It is possible that the hcp structure is more stable than the fcc structure for these MAX phases at 0K or ground state as these fcc phases are found to be transformed back to the hcp phases after long-term annealing treatment. We believe that the fcc phases are probably in a local minimum on the energy landscape since they are all mechanically stable as shown in the reply to comment (c) below.

b) Electronic band structures and phonon dispersion should be calculated and discussed in detail.

In the revision, we additionally calculated electronic density of state (eDOS) of each investigated MAX phase and the results are summarized and discussed in the

supplementary material. Besides, since the supercells of SQS's used in the present work generally do not have fcc symmetry, a complicated mixing method (Ku, *et al. Physical review letters* **104**, 216401 (2010)) is necessary to approximate the electronic band structure and phonon dispersion in the fcc symmetry based on the DFT-calculated charge density and force constants of the SQS's, which is out of the scope of the present work.

Supplementary Figure 5 / Total electronic density of state (eDOS) and projected partial eDOS of the MAX phases. (a) Ti_2AlC ; (b) Ti_3AlC_2 ; (c) Ti_3SiC_2 ; (d) V_2AlC ; (e) Ti_4AlN_3 ; (f) Ti_2AlN ; (g) Nb_4AlC_3 .

The s orbitals of the X elements (i.e. C and N) are generally localized at lower energy states and have an energy gap with the d orbitals of the M elements (i.e. Ti and Nb). In addition, it is found that the s/p orbitals of the A elements (i.e. Al and Si) and the p

orbitals of the X elements (i.e. C and N) are strongly hybridized with the d orbitals of the M elements (i.e. Ti and Nb).

c) For the application in mind, it would probably be interesting (and rather straightforward) to calculate also elastic constants.

In the revised manuscript, additional first-principles calculations were performed to calculate the elastic constants of these fcc MAX phases. The calculations were based on based on the efficient stress-strain method (Shang, *et al. Applied Physics Letters* **90**, (2007); Hu, Y.-J. *et al. Journal of Alloys and Compounds* **671**, 267–275 (2016)). Specifically, independent sets of strains are applied to deform the supercells of the SQSs in the elastic range. For a given set of strains, $\varepsilon_i = (\varepsilon_{i,1}, \varepsilon_{i,2}, \dots, \varepsilon_{i,6})$, lattice

vectors of the strained supercell, \mathbf{R}' , in Cartesian coordinates can be calculated as,

$$\text{Equation 1} \quad \mathbf{R}' = \mathbf{D}\mathbf{R} = \begin{pmatrix} 1 + \varepsilon_{i,1} & \varepsilon_{i,6}/2 & \varepsilon_{i,5}/2 \\ \varepsilon_{i,6}/2 & 1 + \varepsilon_{i,2} & \varepsilon_{i,4}/2 \\ \varepsilon_{i,5}/2 & \varepsilon_{i,4}/2 & 1 + \varepsilon_{i,3} \end{pmatrix} \cdot \begin{pmatrix} a_1 & a_2 & a_3 \\ b_1 & b_2 & b_3 \\ c_1 & c_2 & c_3 \end{pmatrix}$$

where \mathbf{R} are the lattice vectors of the unstrained supercell. The stresses in response to the given strains are calculated via the aforementioned first-principles approach.

Generally, in order to calculate the complete elastic stiffness constants, six independent sets of strains are applied on the structure:

$$\text{Equation 2} \quad \begin{pmatrix} \varepsilon_{1,1} & \varepsilon_{1,6} \\ \varepsilon_{2,1} & \varepsilon_{2,6} \\ \varepsilon_{3,1} & \dots & \varepsilon_{3,6} \\ \varepsilon_{4,1} & \dots & \varepsilon_{4,6} \\ \varepsilon_{5,1} & \varepsilon_{5,6} \\ \varepsilon_{6,1} & \varepsilon_{6,6} \end{pmatrix} = \begin{pmatrix} x & & & & & \\ & x & & & & \\ & & x & & & \\ & & & x & & \\ & & & & x & \\ & & & & & x \end{pmatrix}$$

where $x = \pm 0.01$ in the present work, and the strain components not shown are zero. Using the six sets of corresponding stresses from first-principles, the elastic stiffness constants at a fixed volume, $c_{ij}(V)$, can be determined based on Hooke's law as follows:

$$\text{Equation 3} \quad \begin{pmatrix} c_{11} & c_{12} & c_{13} & c_{14} & c_{15} & c_{16} \\ c_{21} & c_{22} & c_{23} & c_{24} & c_{25} & c_{26} \\ c_{31} & c_{32} & c_{33} & c_{34} & c_{35} & c_{36} \\ c_{41} & c_{42} & c_{43} & c_{44} & c_{45} & c_{46} \\ c_{51} & c_{52} & c_{53} & c_{54} & c_{55} & c_{56} \\ c_{61} & c_{62} & c_{63} & c_{64} & c_{65} & c_{66} \end{pmatrix} = \begin{pmatrix} \sigma_{1,1} & \sigma_{1,6} \\ \sigma_{2,1} & \sigma_{2,6} \\ \sigma_{3,1} & \dots & \sigma_{3,6} \\ \sigma_{4,1} & \dots & \sigma_{4,6} \\ \sigma_{5,1} & \sigma_{5,6} \\ \sigma_{6,1} & \sigma_{6,6} \end{pmatrix} \cdot \begin{pmatrix} \varepsilon_{1,1} & \varepsilon_{1,6} \\ \varepsilon_{2,1} & \varepsilon_{2,6} \\ \varepsilon_{3,1} & \dots & \varepsilon_{3,6} \\ \varepsilon_{4,1} & \dots & \varepsilon_{4,6} \\ \varepsilon_{5,1} & \varepsilon_{5,6} \\ \varepsilon_{6,1} & \varepsilon_{6,6} \end{pmatrix}^{-1}$$

Since the supercells of the SQSs used in the present work do not have a cubic symmetry, the average values of the elastic constants for the conventional cubic structures of the fcc MAX phases were derived using a similar way as employed in the Voigt-Reuss-Hill approach,

$$\text{Equation 4} \quad \overline{C_{11}} = (c_{11} + c_{22} + c_{33})/3$$

$$\overline{C}_{12} = (c_{12} + c_{13} + c_{23})/3$$

$$\overline{C}_{44} = (c_{44} + c_{55} + c_{66})/3$$

where \overline{C}_{11} , \overline{C}_{12} and \overline{C}_{44} are the average for the conventional cubic structures of the fcc MAX phases and the c_{ij} are the elastic constants of the corresponding SQS obtained based on Eq. 3 from the first-principles calculations.

Based on the averaged elastic constants, aggregate properties of the bulk (B), shear (G), and Young's (E) moduli associated with polycrystals were estimated by means of the Voigt approximation (Simmons, *et al.* (M.I.T. Press, 1971)); this provides the upper bound of elastic properties in terms of uniform strains. For a cubic crystal,

Equation 5
$$B = (\overline{C}_{11} + 2\overline{C}_{12})/3$$

Equation 6
$$G = (\overline{C}_{11} - \overline{C}_{12} + 3\overline{C}_{44})/5$$

Equation 7
$$E = 9BG/(3B + G)$$

The calculated elastic constants and moduli of each MAX phase in the fcc structure are summarized in Supplementary Table 4. In addition, all the investigated MAX phases have elastic constants that satisfy the ‘‘Born stability criteria’’ (i.e. $\overline{C}_{11} - \overline{C}_{12} > 0$, $\overline{C}_{11} + 2\overline{C}_{12} > 0$ and $\overline{C}_{44} > 0$) (Mouhat, *et al. Physical Review B* **90**, 224104 (2014).), indicating they are all mechanically stable in the unstressed state in response to the perturbation of elastic strains.

Supplementary Table 4. Calculated elastic constants of the fcc-($M_{n+1}A$) X_n phases. (Unit: GPa)

Phase	\overline{C}_{11}	\overline{C}_{12}	\overline{C}_{44}	B	G	E
fcc-(Ti ₂ Al)C	224.9	89.8	61.3	134.9	63.8	165.3
fcc-(Ti ₂ Al)N	215.8	103.2	68.9	140.7	63.9	166.5
fcc-(V ₂ Al)C	284.8	117.4	70.7	173.3	75.9	198.6
fcc-(Ti ₃ Al)C ₂	264.4	96.5	78.9	152.5	80.9	206.2
fcc-(Ti ₃ Si)C ₂	300.5	115.2	88.8	177.0	90.4	231.6
fcc-(Ti ₄ Al)N ₃	285.1	133.8	109.1	184.2	95.7	244.8
fcc-(Nb ₄ Al)C ₃	296.3	151.8	83.4	200.0	78.9	209.3

5. Question: *The starting structure is not hcp. In the MAX structure, only the MX layers are close-packed. The MAX structure is hexagonal, but not hexagonal close-packed. Please correct everywhere.*

Answer: We agree with the reviewer. While we used the ‘‘hcp’’ designation for consistency with the use of the term in prior work on MAX phases, the structure is not actually fully close-packed. The manuscript has been revised accordingly and the initial MAX phases are now denoted as hexagonal (or ‘‘hex’’ for short) phases.

6. Question: *How is it possible that the structure can remain stable even with 67% C vacancies?*

Answer: Occupancies as low as 12.5% are reported in highly disordered structures, including those produced by irradiation, such as La_2O_3 (Aldebert P., *et al.* Journal de Physique 40.10 (1979): 1005-1012.), Ln_2O_3 (Tracy C., *et al.* Physical Review B 92.17 (2015): 174101.), and so on. Furthermore, if we think about the sites occupied by, for example, interstitial defects in many materials, occupancies of just a few percent are common. For example, the presence of adventitious oxygen in the $4b$ site of UO_{2+x} (Wang J. *et al.*, Scientific reports 4 (2014): 4216.). Additionally, in previous studies, Dridi *et al.* (Dridi Z *et al.* Journal of Physics: Condensed Matter 14, 10237 (2002).) investigated the vacancy effects on structural properties of TiC_x and TiN_x ($0.25 < x < 1$). Based on their conclusion, we believe that the structures with high anion vacancy in our study could exist under irradiation. This is a characteristic of a highly disordered structure.

7. Question: *For the intermediate step, the “gamma” structure should be more clearly explained in words in the text, and a complete description/structural model of the “gamma” structure should (if possible) be provided in the Supplementary Info. The starting MAX structures (from literature) and the resulting fcc structures are provided in the Supplementary information, but the gamma structure is not.*

Answer: The structural model of the $\gamma\text{-(Ti}_3\text{Al)C}_2$ phase is shown in Fig. 1 (c), Supplementary Fig. 1(c), and Supplementary Fig. 3(f) of the revised manuscript. Additionally, the structural information of this phase, including unit cell parameters, site occupancy, and atomic positions, has been added in Supplementary Table 2. The structure of the $\gamma\text{-(Ti}_3\text{Al)C}_2$ phase has been added and described in the supplementary information as well.

8. Question: *Similar disorder occurs in hexagonal intermediate phases $(\text{Cr,Al)Cx}$, $(\text{Ti,Al)Nx}$. These appear similar to the intermediate gamma phase observed here and should be mentioned and discussed: Abdulkadhim, ..., Schneider Surf. Coat. Technol. 206 599 2011 [hexagonal $(\text{Cr,Al)2Cx}$] Cabioch, *et al*, Mater. Res. Bull. 80, 58, 2016 [hcp-like $(\text{Ti,Al)N}$ solid solution]*

Answer: These two solid solution phases that are similar to the intermediate gamma phase in our study have been mentioned and discussed in the revised manuscript.

9. Question: *In the discussion on solid solutions on line 339 and onwards, please make the distinction better between ordered quaternary phases and random solid solutions. $(\text{Cr}_2\text{Ti)AlC}_2$ is a quaternary phase – or ordered solid solution depending on how you view it - with a nominally fixed stoichiometry [see also B. Anasori *et al*, Journal of Applied Physics 118:94304-7 (2015) for a more detailed discussion on this phenomenon]. The other example is a regular solid solution, which is a very different thing. Also, a detail for the carbonitrides: probably a better (=more original from the same authors) reference than ref 47 is Cabioch *et al* J. Eur. Ceram. Soc. 32 1803 2012, and the original is M.W. Barsoum, M. Ali, T. El-Raghy Metall Mater Trans A, 31 (2000), pp. 1857-1865*

Answer: We agree with the reviewer. Accordingly, we have differentiated the ordered quaternary phases from the random solid solution phases in the revised manuscript and cited the more appropriate references.

10. Question: *There are no CoI or Data Availability Statements. For the latter, it is highly recommended that the authors make all data and files available in a public data repository (eg Zenodo) after acceptance.*

Answer: As recommended, we will make them available online if it is accepted.

11. Question: *Abstract: “A grand challenge in complex ceramic materials is to understand the nature of their structural disordering” Please correct to “A challenge...”. A “grand challenge” is an established concept; it is much more than a research problem, it is a major undertaking with global impact, something like for example stopping global warming. (en.wikipedia.org/wiki/Grand_Challenges)*

Answer: The word “grand” has been deleted from the revised manuscript.

12. Question: *Materials and methods: “Details of the synthesis progress have been published elsewhere.” Please provide references.*

Answer: The reference for the details of the synthesis process was added in the revised manuscript.

13. Question: *The paper is well written but should be carefully proofread for spelling and mixed up words. For example, lines 134-137: “...edge-sharing Ti6C octahedral (SHOULD BE “OCTAHEDRA”, NO L) interleaved by close-packed Al layers, with Ti atoms in 2a and 4f, Al atoms in 2b, and C atoms in 4f Wyckoff positions. The intermeditated (SHOULD BE “INNER”) Ti layer is donated (SHOULD BE “DENOTED”) as Ti(I) layer (REMOVE “LAYER”), while Ti layers...” Etc.*

Answer: All such mistakes in the manuscript have been revised.

14. Question: *Line 95: “In the absent direct observation...” should be “In the absence of direct observation...”*

Answer: The manuscript has been revised according to the reviewer’s suggestion.

15. Question: *The title should rather be “Disorder...” than “Disordering...”*

Answer: The title has been revised according to the reviewer’s suggestion.

Reviewer #2 (Remarks to the Author):

1. Question: *page 6 line 116-117: I do not understand the $(n+1)/n$ ratio; the proportion of M atoms on the cation site should be $(n+1)/(n+2)$. In the manuscript, the anion site occupancy is 50% (I agree with this value) whereas the value given in line 117 is $n/(2n+1)$.*

Answer: We apologize for our careless mistakes. The reviewer is right. The ratio of M and A cations is $(n+1):1$, and the X anions are located at the anion sites with an occupancy of $n/(n+2)$. The manuscript has been revised to correct the ratio and the occupancy of the anions.

2. Question: *page 7 line 136: Change “donated” by denoted.*

Answer: We apologize for our careless mistake. The manuscript has been revised according to the reviewer’s suggestion.

3. Question: *page 8 line 161-164: There is, to my opinion, a mistake. $Al_{Ti(II)}$ antisite defects in Ti_3AlC_2 exhibits the lowest formation energy (0.74 eV) among all defect types (1.65 eV for $Al_{Ti(I)}$) which is indeed consistent with your results.*

Answer: We apologize for our careless mistake. As the reviewer mentioned, the formation energy of the $Al_{Ti(I)}$ antisite defect is indeed 1.65 eV and that of $Al_{Ti(II)}$ is 0.74 eV. The manuscript has been revised to correct this mistake in accordance with the reviewer’s suggestion.

4. Question: *It is quite important to have an idea of the sample thickness and to know if:*

- the different samples have about the same thickness

- the correct thickness has been used in simulated STEM HAADF images.

Answer: Thank you for your suggestion. As recommended, we have employed TEM EELS to measure the thickness of all the Ti_3AlC_2 samples within an experimental error of $\pm 10\%$, as shown in the figure below. Based on the EELS spectrum, the thickness values of all these samples (the pristine sample, and the samples irradiated at 3×10^{14} and 2×10^{16} cm^{-2} , respectively) were determined (~ 12 nm, ~ 18 nm, and ~ 13 nm) using the “compute thickness” function in the DigitalMicrograph 3 software. These values differ slightly and the sample irradiated at 3×10^{14} cm^{-2} is thicker than the others. However, considering the error generated in the experiments, the effect of the thickness difference should not be major. In addition, for the STEM image simulation, the thickness of the sample was fixed at the values measured here.

Electron Energy Loss Spectrum of the pristine sample and the samples irradiated at 3×10^{14} and $2 \times 10^{16} \text{ cm}^{-2}$, respectively.

5. Question: Page 13 line 262-264: The sentence “This is attributed tophase.” has to be rewritten.

Answer: The sentence has been revised to “This is attributed to the facts that the Ti(II)-Al bonding is the weakest among all bonds (such as the strongly covalent Ti-C) in the system and the Ti(II)-Al-Ti(II) distance is the largest in the Ti_3AlC_2 phase.”

6. Question: The authors mention, in the conclusion (page 17 lines 364-367), that ion irradiation allows phase transformations to otherwise unachievable solid solution phases. It can lead to new derivatives of the MAX phases with tailored disorder and potentially improved properties. It has to be moderated as the size of the solid solution sample is necessary very small (typically the size of the damage region which should be about 400 nm for irradiation performed with Au^+ at 1 MeV).

Answer: Irradiation with 1 MeV Au ions can only modify the structure of the region with a depth of several hundreds of nanometers. However, the use of higher ion energies, such as in the case of swift heavy ions, might trigger the same transformation in larger volumes. we have revised the sentence to “solid solution phases, which can provide a new strategy for the design of new derivatives of the $\text{M}_{n+1}\text{AX}_n$ phases with tailored disorder and potentially improved properties within the typically thin ion-solid interaction region.”

7. Question: The reader (and the referee) would appreciate to know if the quality of the STEM HAADF and ABF images was as good as the presented ones for a lot of sample regions. Indeed, irradiation introduces damages and it is very often quite difficult to get good quality images due to this damages.

Answer: In our work, irradiation-induced structural damages were also observed in our STEM images. Lattice distortion induced by small defect clusters can be also found during the phase transformation process, especially when at least two phases coexist. However, because this manuscript is focused on the irradiation-induced new structures in the MAX phases, we have sought to exhibit the evidence and data that

most clearly illustrates these new structures and to avoid the interference of irradiation-induced structural damage. Thus, collected images that, due to the effects of the defect concentration, did not clearly demonstrate to the reader the transformations observed were not included in the final manuscript.

8. Question: *The damage profile should be given in order to get an idea of the depth of the damage region.*

Answer: The depth profiles of the damage level and implanted ion concentrations induced by 1 MeV Au⁺ ion irradiation have been added in the supplementary information accordingly.

Supplementary Figure 7 | Damage profile in Ti₃AlC₂. Depth profiles of damage level, expressed in displacements per atom (dpa), and implanted ion concentration induced by 1 MeV Au⁺ ion irradiation at $1 \times 10^{16} \text{ cm}^{-2}$ in Ti₃AlC₂. These data were calculated using the SRIM code.

9. Question: *It would be of interest to get the displacement per atom (dpa) for the different fluences that have been used and to have an idea of the dpa at the observed region (figure 1g, 1j and 1m).*

Answer: According to the suggestion, the displacement per atom for different fluences, calculated using the SRIM code, have been added to the revised manuscript.

10. Question: *Page 23 line 467: change “donated” by denoted*

Answer: We apologize for our careless mistakes. The manuscript has been revised according to the reviewer’s suggestion.

11. Question: *Figure 1j and 1s are of quite poor quality. Is the sample thickness higher than the ones that allows obtaining the other images (1g, 1p and 1m, 1s)? If it is not the case, is it the result of irradiation?*

Answer: The quality of STEM images can be influenced by many factors, including the condition of the TEM, the quality of TEM specimens, the parameters used for images and so on. We used the same method (i.e. condition of the TEM facility, etc.) to prepare the samples and perform the STEM imaging. As the reviewer mentioned, the thickness is definitely a key factor when analyzing the results, although we controlled the experimental conditions to the extent possible. The thickness of the sample irradiated at $3 \times 10^{14} \text{ cm}^{-2}$ (shown in Fig. 1 j, s) is $\sim 18 \text{ nm}$, which is indeed a little larger than those of the other two samples ($\sim 12 \text{ nm}$, and $\sim 13 \text{ nm}$, respectively). However, for the STEM experiments in our study, we focused on the edge region in each TEM specimen such that we could make sure the observed regions are as thin as possible. This small difference of thickness might slightly influence the quality of the images. Additionally, irradiation can result in the formation of various defects, which can cause lattice distortion and break the local order, influencing image quality. Especially, as antisite defects break the initial arrangement of cations, there occurs a large amount of disorder. Therefore, we speculate that the lower quality of these STEM images is attributed to the irradiation-induced structural damage.

12. Question: *I am surprised that Figure 2 represents simulated electron diffraction patterns. I would say that it is experimental diffraction patterns.*

Answer: We apologize for this mistake. Assuming that this is a reference to Supplementary Figure 2, these are indeed the experimental diffraction patterns. The caption has been revised accordingly.

Reviewer #3 (Remarks to the Author):

1. Question: *Have the authors evaluated the distribution of injected Au⁺ ions in samples and their role in stabilizing the fcc-structured (M_{n+1}A)X_n phases?*

Answer: Thank you for your comment. When performing the STEM experiments, we focused on the peak damage region, such that the region containing the maximum concentration of the deposited Au ions was avoided (Supplementary Fig. 7). We also revised the manuscript to clarify this point. According to SRIM calculation, the Au ion concentration in this region at the fluence of 2×10^{16} cm⁻² is only ~ 0.8 %. Considering the relative chemical inertness of deposited Au ions, their low concentration, and the low migration motion (room temperature irradiation), we conclude that the influence of deposited Au atoms to the new structure is minor. In fact, Au ion irradiation has been widely used for studying irradiation-induced structural damage in ceramics due to its chemical inertness and high damage efficiency, including ZrO₂ (Zhang, *et al.* NIMB, 338 (2014): 19-30.), SiC (Jiang *et al.* Physical Review B 80.16 (2009): 161301.), ZrC (Gosset, D., *et al.* Journal of nuclear materials 373.1-3 (2008): 123-129.) and MAX phases (Wang *et al.* Acta Materialia 98 (2015): 197-205).

2. Question: *In electron microscopy, electron beam irradiation usually introduces defects in specimen and may lead to local ordered - disordered transformation. Did the authors evaluate the effects of electron irradiation during STEM measurements?*

Answer: In this study, the electron energy was 200 keV, which might introduce some defects in ceramic materials (Huang *et al.* Scripta Materialia 113 (2016) 114–117). However, in our study, we did not observe any structural modification induced by electron irradiation. The structures of the initial hexagonal phase and the intermediate γ phase and fcc phase did not change during the experiment, indicating that these phases are quite stable when exposed to the electron beam. Therefore, we believe that any significant effect of electron beam exposure on our results can be excluded.

3. Question: *Perfect electron microscopy work, it's impressive.*

Answer: Thank you so much for this comment.

REVIEWERS' COMMENTS:

Reviewer #1 (Remarks to the Author):

The authors have thoroughly addressed all comments from the three reviewers and performed the necessary additional Rietveld refinement and computational work. I am pleased to see that the other two reviewers share my positive view of the manuscript.

In the new supplementary information: on p.13, full details on the refinement (not just the end result) should be provided, following standard procedure for presenting Rietveld refinement.

Once this amendment has been made, I am pleased to recommend this paper for publication in Nature Communications.

Best regards,
Per Eklund
(signed review)

Reviewer #2 (Remarks to the Author):

The originality of the approach (the use of HAADF and ABF techniques to image the disorder) is clearly explained and is convincing. To my opinion, the results are of main interest for the MAX phase community and also for researchers that are working on order-disorder transition and for nuclear field.

The paper is of high quality level and its quality has increased thanks to the work of all referees. It can now be published.

Sylvain Dubois

Reviewer #3 (Remarks to the Author):

The authors have answered all my doubts and I have no more questions.

Responses to reviewer:

Thank you very much for the kind and detailed comments that have helped us to bring clarity to the presentation of our results. Please find our answers to the specific issues raised:

Reviewer #1 (Remarks to the Author):

The authors have thoroughly addressed all comments from the three reviewers and performed the necessary additional Rietveld refinement and computational work. I am pleased to see that the other two reviewers share my positive view of the manuscript. In the new supplementary information: on p.13, full details on the refinement (not just the end result) should be provided, following standard procedure for presenting Rietveld refinement. Once this amendment has been made, I am pleased to recommend this paper for publication in Nature Communications.

Answer: As suggested, we have provided the more detailed information of the Rietveld refinement in the Supplementary information.

Reviewer #2 (Remarks to the Author):

The originality of the approach (the use of HAADF and ABF techniques to image the disorder) is clearly explained and is convincing. To my opinion, the results are of main interest for the MAX phase community and also for researchers that are working on order-disorder transition and for nuclear field. The paper is of high quality level and its quality has increased thanks to the work of all referees. It can now be published.

Answer: Thank you very much for your endorsement for publication.

Reviewer #3 (Remarks to the Author):

The authors have answered all my doubts and I have no more questions.

Answer: Thank you.